

# Intergenic incompatibilities reduce fitness in hybrids of extremely closely related bacteriophages

Andrew M. Sackman, Danielle Reed and Darin R. Rokyta

Department of Biological Science, Florida State University, Tallahassee, FL,
United States of America

## ABSTRACT

Horizontal gene transfer and recombination occur across many groups of viruses and play key roles in important viral processes such as host-range expansion and immune-system avoidance. To have any predictive power regarding the ability of viruses to readily recombine, we must determine the extent to which epistasis restricts the success of recombinants, particularly as it relates to the genetic divergence between parental strains. In any hybridization event, the evolutionary success or failure of hybrids is largely determined by the pervasiveness of epistasis in the parental genomes. Recombination has previously been shown to incur steep fitness costs in highly divergent viruses as a result of disrupted epistatic interactions. We used a pair of bacteriophages of the family Microviridae to demonstrate that epistasis may evidence itself in the form of fitness costs even in the case of the exchange of alleles at a locus with amino acid divergence as low as 1%. We explored a possible biophysical source of epistasis in the interaction of viral coat and scaffolding proteins and examined a recovery mutation that likely repairs interactions disrupted by recombination.

## INTRODUCTION

Most evolutionary processes, from adaptation to speciation, are influenced by the presence of epistasis, or interactions between genetic loci. Epistasis has been implicated in the evolution of sexual reproduction (*Otto & Gerstein, 2006*), and an understanding of epistasis is essential to the field of personal genomics and to the unraveling of the genetic architectures of complex diseases (*Moore & Williams, 2009*; *Green et al., 2010*). Epistasis also poses a potential barrier to the evolutionary success of recombinant genomes (*Kouyos, Silander & Bonhoeffer, 2007*; *Rokyta & Wichman, 2009*; *Sackman & Rokyta, 2013*; *Doore & Fane, 2015*), and genic incompatibility, such as may be caused by epistasis, can enforce divergence between distinct lineages (*Coyne & Orr, 2004*). Horizontal gene transfer and recombination occur naturally in a wide array of viruses, including hepatitis E virus (*Wang et al., 2010*), hepatitis B virus (*Lyons et al., 2012*), and humam immunodeficiency virus (*Motomura, Chen & Hu, 2008*; *Rigby et al., 2009*; *Ssemwanga et al., 2011*), as well as in microvirid bacteriophages (*Rokyta et al., 2006*), and these processes

Corresponding author
Darin R. Rokyta,
drokyta@bio.fsu.edu

may play a large role in microbial evolution and divergence (*De la Cruz & Davis, 2000*; *Rokyta et al., 2006*). *Botstein (1980)* proposed that the genomes of bacteriophages are in large part a mosaic of interchangeable genetic elements that are regularly exchanged through recombination with other bacteriophage genomes of varying degrees of relation, an idea supported by significant evidence from nature (*Hendrix et al., 1999*; *Hendrix, 2002*; *Hatfull, Cresawn & Hendrix, 2008*). The success of recombinant viral genomes requires either compatibility between genotypes or that any incompatibility be weak or easily surmountable by means of compensatory evolution.

The reduced fitness that may handicap most potential recombinants as a result of epistatic incompatibility may not be easily observed in a natural setting, where inviable or unfit hybrids will be quickly lost. However, several experiments have demonstrated the high degree of divergence that may be tolerated between ancestral genomes during recombination events. *Kacharoo et al. (2015)* demonstrated that nearly half of all essential yeast genes could be replaced with human orthologs while still producing viable yeast cells. Homologous recombination between bacteriophages at loci as divergent as 8% (*Rokyta & Wichman, 2009*), 22% (*Springman et al., 2012*), and even ≥60% (*Sackman & Rokyta, 2013*; *Doore & Fane, 2015*) at the amino acid level have been performed experimentally and produced viable, but extremely sick, hybrid phages. *Rokyta & Wichman (2009)* successfully conducted a reciprocal cross of the gene encoding the coat protein (F) between microvirid bacteriophages ID12 and ID2 (*Rokyta et al., 2006*). Two viral species of the family *Geminiviridae* that were 18% divergent at the nucleotide level were recombined by means of random DNA shuffling, and all of the resulting recombinants were able to successfully infect their hosts (*Vuillaume et al., 2011*). Alleles from bacteriophages $\phi$X174 and G4 (which are divergent at 33% of their nucleotide sites genome-wide) can, in some cases, complement each other and facilitate rescue of mutant genotypes with a single defective protein during coinfection (*Borrias et al., 1979*), and *Doore & Fane (2015)* successfully crossed $\phi$X174 and G4 at the major spike protein (G). *Springman et al. (2012)* suggested that the maximum level of divergence that may be tolerated in recombination events may be complex and vary in each individual case.

Although recombination may yield viable chimeric viruses, hybridization generally entails high fitness costs resulting from epistasis that could preclude the persistence of hybrids in wild populations containing high-fitness competitors. *Rokyta & Wichman (2009)* showed that the insertion of an allele of the microvirid phage gene F into a background to which it was not adapted resulted in recombinants with fitnesses less than half that of their background ancestral genotype. Fitness was, however, recoverable with subsequent compensatory evolution. Furthermore, fitness recovery was swift and simple, requiring only five or fewer mutations, with the first mutation to fix in each recovery population accounting for approximately 60% of total fitness recovery, on average. The phage genotypes used in that experiment were highly divergent, differing at 19% of nucleotide sites over the whole genome, excluding gaps. At the amino acid level, the coat proteins (F) are approximately 8% different (*Rokyta & Wichman, 2009*). In a similar experiment, *Doore & Fane (2015)* exchanged the spike protein (G) between highly

divergent microviruses $\phi$X174 and G4, which share an amino acid similarity of only 40% between their spike proteins. *Doore & Fane (2015)* found that, although one of the two resulting hybrids was viable and able to recover some lost fitness with a single adaptive mutation, the second chimera was only viable in the presence of an exogenous supply of the original spike and DNA pilot (H) proteins. This hybrid, not viable without ancestral copies of the spike and pilot proteins provided in the growth environment, required additional mutational steps for fitness recovery, first globally decreasing gene expression prior to modifying key protein-protein interactions, and finally re-elevating gene expression. *Doore & Fane (2015)* hypothesized that the presence of the foreign spike protein improved the thermodynamic or kinetic favorability of non-productive reactions that hinder assembly of the capsid, demonstrating a mechanism by which disrupted protein-protein interactions resulting from horizontal gene transfer may reduce or altogether eliminate the viability of newly created hybrids. These cases highlight the ability of highly divergent genomes to hybridize and overcome high fitness costs resulting from disrupted epistatic interactions, given sufficient time and resources to allow compensatory evolution. Whether similar epistatic barriers to recombination are present even between highly similar genomes remains to be determined. The level of genetic divergence at which epistatic barriers to hybridization emerge determines the evolutionary distance beyond which related populations may no longer successfully hybridize and is an important threshold in the process of speciation.

We performed an experiment similar to that of *Rokyta & Wichman (2009)*, conducting a reciprocal cross of two microvirid phage genomes at coat protein gene F. Unlike the pair of bacteriophages used in this previously described experiment, our selected phages, ID12 and NC6 (*Rokyta et al., 2006*), were extremely similar. Their genomes are of identical length (5,529 nucleotides; *Rokyta et al., 2006*), and these phages are divergent at only 7% of nucleotide sites across their genomes. They are extremely similar in the region of the coat protein gene, differing at only five amino acid sites of the 427 sites comprising the protein. The other gene products composing the mature phage capsid and its scaffolding proteins differ at the amino acid level as follows: the spike protein (G), 14%; the pilot protein (H), 7%; the DNA-binding protein (J), 0%; the internal scaffolding protein (B), 8%; and the external scaffolding protein (D), 1%. The two phage genomes display a remarkable level of similarity relative to other members of the family *Microviridae*, especially in gene F. The pair of genotypes used by *Rokyta & Wichman (2009)*, ID2 and ID12, were the two most distantly related phages that could be made to produce viable recombinants at the F locus under standard laboratory conditions. Alternatively, ID12 and NC6 represented the two most closely related phages which have been adapted to our experimental growth conditions (described in 'Methods') by *Rokyta, Abdo & Wichman (2009)*. We addressed whether epistasis may evidence itself in the form of fitness costs in instances of recombination between phages with low divergence and, if so, how the magnitudes of those costs compare to cases of high-divergence hybridization. We also investigated whether recovery of any lost fitness in hybrids was as swift and simple as

observed in cases where divergence was high, and we sought to identify the causes of any epistatic incompatibilities in these hybrid bacteriophages.

## METHODS

### Ancestral bacteriophage genotypes

ID12 and NC6 were originally isolated and described by *Rokyta et al. (2006)*. Wild-type isolates of ID12 and NC6 were adapted to culture conditions via serial flask transfer for 60 and 70 passages, respectively (*Rokyta, Abdo & Wichman, 2009*). The GenBank accession numbers of the original isolates are DQ079905 (ID12) and DQ079907 (NC6). A random isolate from the adapted population ID12a60 was used as the ID12 ancestor for genome recombination. Full genome sequencing confirmed that the isolate possessed both of the mutations that fixed in the ID12a lineage of *Rokyta, Abdo & Wichman (2009)*: 1970 (A → G) and 4919 (G → A). The selected isolate of ID12 had an additional mutation at position 4843 (A → G) previously found to be present in half of the sequenced isolates of the ID12a60 population (*Rokyta & Wichman, 2009*). The isolate used as our NC6 ancestor was selected at random from the NC6a70 adapted population. This isolate contained all of the mutations that fixed during adaptation of the NC6a lineage: 2418 (C → T), 4168 (A→ G), 5033 (G → A), 5071 (G → A) (*Rokyta, Abdo & Wichman, 2009*).

### Construction of hybrid phage genotypes

Recombination was accomplished using the technique previously described by *Rokyta & Wichman (2009)*. A set of primers in both directions flanking the region containing coat protein gene F (positions 2552–3835 in ID12 and NC6) was created for both ancestral genotypes, for a total of eight primers. The arrangement of primer pairs allowed amplification of gene F from each donor phage and amplification of the remainder of the genome from each recipient phage. The primers were designed to consist of roughly half ID12 and half NC6 sequence, containing the desired hybrid phage sequence and centered around the stop or start codon of gene F. Donor and recipient fragments were amplified separately in a standard polymerase chain reaction (PCR), and then combined in approximately equal copy numbers in a PCR without primers to assemble the complete genome. The recombinant PCR products were purified, electroporated into *Escherichia coli* strain C, and plated. Isolates from the resulting plaques were confirmed to be recombinant and free of additional mutations by full genome sequencing. Using this procedure, we created chimeric genomes containing gene F from NC6 in an ID12 background (ID12-NC6F) and ID12 gene F in an NC6 background (NC6-ID12F).

Site-directed mutagenesis (*Pepin, Samuel & Wichman, 2006*; *Pepin & Wichman, 2007*) was used to create the ID12Mut genotype, consisting of the ancestral ID12 genome with the mutation observed in the ID12-NC6F mutation during fitness recovery, to confirm that the mutation that fixed in the ID12-NC6F recovery line was beneficial only in the context of the hybrid background. A set of primers centered on the mutation at site 3485 (A → G) containing the desired base state was used with pre-existing primers to generate two amplified genome fragments overlapping at the mutation site and at a segment of DNA

located on the opposite end of the circular genome. The amplified genome fragments were combined in a PCR without primers to assemble the complete genome. The mutagenesis product was purified, electroporated, isolated, and confirmed to be free of additional mutations by full-genome Sanger sequencing.

### Fitness assays and recovery

Fitness assays were performed as described by *Rokyta & Wichman (2009)*. Fitness was measured as the $\log_2$ increase in total phage per hour. The phage host, *E. coli* C, was grown to a concentration of $1-2 \times 10^8$ cells per ml in phage Lysogeny broth (10 g NaCl, 10 g tryptone, and 5 g yeast extract per liter) supplemented with 2 mM CaCl$_2$ in 125-ml flasks at 37 °C shaking at 200 rpm in an orbital water bath. Approximately $10^4-10^5$ phage were added and grown for 40 min. Growth was terminated with chloroform. Fitness was measured in 10 replicates for each genotype. Fitness recovery of the ID12-NC6F and NC6-ID12F lines was performed through serial flask transfers under conditions similar to fitness assays with approximately a 10-fold increase in the number of phage added to each flask. Control lines using a population grown from the ancestral ID12 and NC6 isolates were run for the same number of growth periods as the ID12-NC6F and NC6-ID12F recovery lines to confirm that any increase in fitness observed in the hybrids was contingent upon recombination. All statistical analyses of fitness values were performed with R (*R Development Core Team, 2010*).

## RESULTS AND DISCUSSION

### An asymmetrical cost of hybridization

The isolates of ID12 and NC6 used as ancestors to create the recombinant phages came from populations that had previously been adapted to our culture conditions by *Rokyta, Abdo & Wichman (2009)* until fitness plateaued and remained stable for at least 20 growth periods (approximately 60 generations). Both ID12 and NC6 had high fitnesses, showing $23.4 \pm 0.45$ and $26.7 \pm 0.30$ doublings per hour, respectively ($\pm$indicates standard error). We performed a reciprocal exchange of the two alleles for the coat protein gene (F). The coat protein is present in both the procapsid and mature phage capsid and interacts directly with all of the phage structural and scaffolding proteins during capsid assembly, making it ideal for detecting epistasis by means of experimental hybridization. Replacement of the allele for gene F in the ID12 genome by the homolog from NC6 resulted in a phage, ID12-NC6F, with fitness $23.7 \pm 0.33$, slightly higher than the fitness of its primary ancestor, ID12, but not significantly so ($t$-test, two-sided, unequal variance, $P = 0.50$; Fig. 1). The replacement of NC6 gene F with the homolog from ID12, yielding genotype NC6-ID12F, resulted in a phage with fitness $25.5 \pm 0.23$, a significant decrease of 1.2 doublings per hour from the fitness of its primary ancestor, NC6 ($t$-test, two-sided, unequal variance, $P = 0.01$; Fig. 1).

Recombinants of the closely related phages ID12 and NC6 did not suffer the same staggering fitness costs observed in recombinants of ID2 and ID12 involving the same gene, which showed an average fitness decrease of 13.9 doublings per hour relative to ancestral fitness (*Rokyta & Wichman, 2009*). Prior to fixation of several compensatory

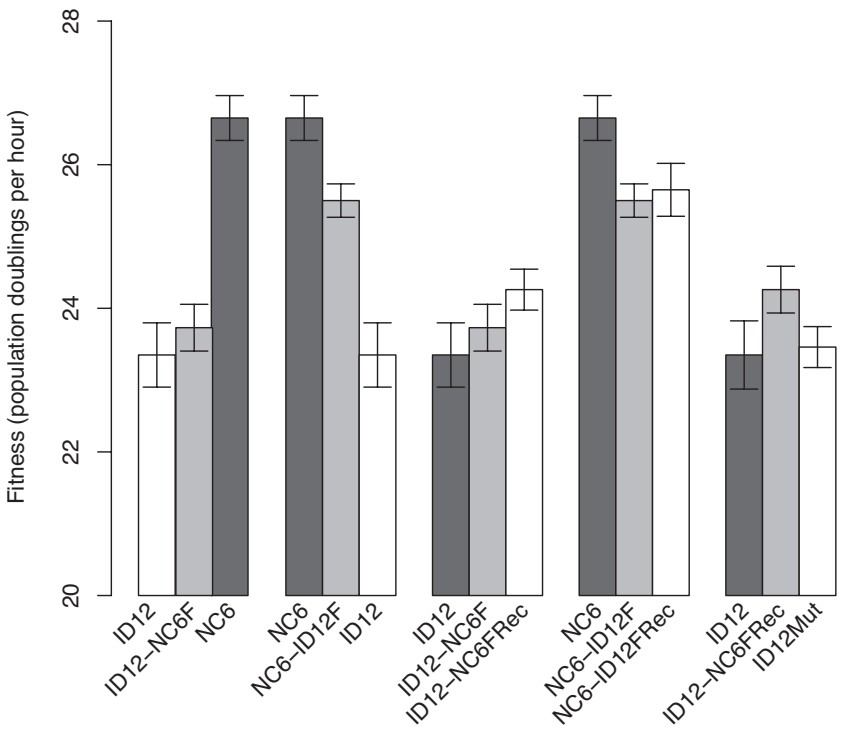

**Figure 1 Comparisons of phage fitnesses resulting from hybridization and subsequent compensatory evolution showed evidence for hybrid incompatibilities between close relatives.** ID12 and NC6 were the ancestral genotypes used to produce the ID12-NC6F hybrid (ID12 genome with NC6 F allele) and NC6-ID12F hybrid (ID12 F allele in NC6 background). ID12-NC6FRec and NC6-ID12FRec are measurements from the final passage of the recovery population of each hybrid, and ID12Mut is an isolate containing the mutation that fixed in the ID12-NC6F recovery population at nucleotide site 3485 placed in the ID12 background. The first two fitness comparisons demonstrate the asymmetrical effects of hybridization at gene F between ID12 and NC6. The third and fourth comparisons demonstrate the positive, but statistically insignificant, fitness effects of the first mutation to fix in each recovery lineage, and the final comparison shows the neutrality of the recovery mutation that fixed in the ID12-NC6F recovery line in the context of the original F allele.

mutations, the ID2 and ID12 hybrids were not capable of outcompeting either parent and would likely have been quickly eliminated in any natural environment with competition for hosts. Our NC6 and ID12 hybrids, by contrast, had intermediate fitnesses relative to their two ancestors, and could easily have sustained populations within a larger community containing their parents. This potential difference in long-term hybrid viability implies that recombination between more closely related strains is likely to have evolutionarily important consequences under a wider array of conditions than recombination between highly divergent strains.

ID12-NC6F showed a slightly higher, but not significantly so, fitness than its background ancestor. NC6-ID12F, however, did show a significantly lower fitness than its NC6 ancestor. In one case, hybridization yielded a very slight and non-significant fitness benefit, and the reciprocal case yielded a significant fitness cost. In the absence of epistasis, we would expect the fitness effect of a reciprocal cross to have similar magnitude and opposite sign in the two recombinants. The asymmetry of the observed fitness effects strongly

suggested the presence of epistasis and indicated that the fitness of both recombinants may have been lower than should be expected in the absence of epistasis, although the effects of epistasis were weaker than observed by *Rokyta & Wichman (2009)*. This evidence indicates that recombination was more favorable between closely related genotypes, but that epistasis was still present and consequential, even with low divergence between genotypes.

## Hybrid incompatibilities impact fitness

The major difficulty in testing for epistasis in the context of our experiments is determining the expected additive effects of hybridization. The expected additive effect of exchanging the coat protein gene F is contingent upon the proportion of overall fitness determined by variation in gene F. To account for this, we tested for and successfully demonstrated epistasis by looking for violations of null hypotheses for which gene F contributed to overall genome fitness at four different levels and for which intergenic epistatic interactions were absent. Our null hypotheses included: (1) variation in gene F had no effect on fitness, (2) variation in gene F was the sole determinant of fitness, (3) recombination of homologous regions of the genome resulted in a new genotype fitness that was the weighted average of the two ancestral fitnesses, proportional to the fraction of each genome composing the novel genotype (as would be the case where variation at each site in the genome contributes equally to overall fitness), and (4) variation in gene F accounts for 50% of the fitness of a genotype (as would be the case where variation in F is a relatively important determinant of overall fitness). Both ancestral phages were pre-adapted to, and highly fit in, their environment, and both alleles of capsid protein gene F were highly adapted in the ancestral backgrounds. Any deviation from expected fitness values should therefore reflect only the effects of disrupting favorable epistatic interactions in the ancestral genomes.

The first two null hypotheses we tested assumed that epistasis was absent and that overall fitness was determined either not at all or entirely by variation in gene F. In the case where F was not a determinant of fitness, recombination should have had no effect on fitness, and a hybrid phage would retain the same fitness as its non-F ancestor. The fitness of ID12-NC6F was nearly identical to expected fitness under this condition, and the fitness of NC6-ID12F was significantly less than expected (Table 1). In the second case, where F determines 100% of fitness, the fitness of a hybrid was expected to be equal to the fitness of the ancestor from which its F allele was derived. In this case, the actual fitness of ID12-NC6F was significantly lower than the expected recombinant fitness, and the fitness of NC6-ID12F was significantly higher than the expected value (Table 1). In both of these hypothetical cases representing the theoretical extremes of how variation in gene F may contribute to overall fitness, our results indicate that epistasis was acting significantly to reduce fitness below expectations within at least one of the two hybrid genotypes.

If we assume that variation at each site in the genome was of equal importance in determining overall fitness, then recombination of homologous regions of the genome should result in new genotype fitnesses that were an average of the two ancestral fitnesses, proportional to the percent of the hybrid genome derived from each ancestor. In this
**Table 1 Tests of null expectations of hybrid fitness.** Expected and observed fitness and $t$-test results of both hybrid genotypes under each of four sets of fitness expectations: (1) fitness is not at all determined by variation in F; (2) fitness is entirely determined by variation in F; (3) fitness is determined equally by variation at each site in the genome; (4) half of all fitness is determined by variation in F. At least one genotype significantly diverges from expected fitness under three of four null hypotheses.

| Hypothesis | Fitness determinant | Genotype | $w_E$ | $w_O$ | $p$ value |
|---|---|---|---|---|---|
| 1 | Genome, not F | ID12-NC6F | $23.4 \pm 0.45$ | $23.7 \pm 0.33$ | 0.5 |
| | | NC6-ID12F | $26.7 \pm 0.30$ | $25.5 \pm 0.23$ | <0.001 |
| 2 | F, not genome | ID12-NC6F | $26.7 \pm 0.30$ | $23.7 \pm 0.33$ | $<10^{-5}$ |
| | | NC6-ID12F | $23.4 \pm 0.45$ | $25.5 \pm 0.23$ | < 0.001 |
| 3 | Proportional to length | ID12-NC6F | $24.1 \pm 0.44$ | $23.7 \pm 0.33$ | 0.55 |
| | | NC6-ID12F | $25.9 \pm 0.41$ | $25.5 \pm 0.23$ | 0.51 |
| 4 | 1/2 genome, 1/2 F | ID12-NC6F | $25.0 \pm 0.46$ | $23.7 \pm 0.33$ | <0.05 |
| | | NC6-ID12F | $25.0 \pm 0.46$ | $25.5 \pm 0.23$ | 0.33 |

case, absent epistasis, the substitution of genomic material from a higher fitness genotype should increase fitness, and substitution of sequence from a lower fitness ancestor should result in decreased fitness. Under this expectation, fitnesses of both hybrids were lower than expected, but not significantly so (Table 1). We therefore cannot reject a lack of epistasis if we assume that each site in the phage genome contributes equally to fitness.

Empirical evidence suggests that variation in gene F contributes more to overall fitness than in other genes. *Rokyta et al. (2005)* performed replicate adaptations of a bacteriophage (ID11) closely related to our two ancestral genotypes to lab conditions and found that seven of nine first-step mutations, where the first mutation to fix in an adapting population generally accounts for the greatest fitness gain during adaptation, affected sites in gene F. *Miller, Joyce & Wichman (2011)* performed six replicate adaptations of ID11, and, of 14 unique first step mutations observed, ten affected sites in gene F. Four out of 14 unique mutations observed in the recovery lines of ID2 and ID12 hybrids also occurred in gene F (*Rokyta & Wichman, 2009*), and two of three mutations observed in a line of $\phi$X174 adapted to high temperatures were in gene F (*Bull, Badgett & HA, 2000*). These empirical results suggesting that variation in F was a relatively large determinant of overall fitness make logical sense considering that the coat protein is the centerpiece of the capsid assembly reaction, which determines growth rate and, therefore, fitness. If we assume that gene F contributes 50% of total phage fitness and epistasis was absent (based on the evidence suggesting that F may be responsible for a larger share of fitness determination than the percentage of sites it constitutes), then replacing an F allele with a homolog from another genotype should have resulted in a hybrid genome with a fitness that was an exact average of the fitness of both ancestors. The observed fitness of ID12-NC6F was significantly lower than this null expectation, and the fitness of NC6-ID12F did not differ significantly from expected fitness (Table 1). At least one of two hybrids (ID12-NC6F) showed a significantly lower than expected fitness following the replacement of F with an allele from a higher fitness ancestor, suggesting the presence of epistasis.

Gene F most likely accounts for at least a portion of overall fitness proportional to the percent of the genome it occupies and probably an even greater portion given the amount of empirical evidence suggesting that mutations in gene F were critical to fitness gains during adaptation. Disregarding epistasis, the replacement of the F allele in ID12 with an allele from a phage of higher fitness should have resulted in a hybrid with increased fitness relative to ID12, but the fitness of ID12-NC6F was not significantly different from the fitness of ID12. The expected benefit of a higher fitness allele appears to be lessened by the disruption of favorable epistatic interactions between gene F and the rest of the genome. The fitness of ID12-NC6F was significantly lower than expected under two sets of null expectations and provided clear evidence of intergenic epistasis. In addition, the fitness of NC6-ID12F was lower, albeit not significantly, than expected under null hypothesis three, suggesting that epistasis may also result in a lower fitness than can be accounted for solely by the substitution of a lower fitness F allele into the NC6 genome.

Regardless of the precise contribution of gene F to overall fitness or the additive expectations in this experiment, the asymmetry of the fitness effects, with no significant change in ID12-NC6F and a significant cost observed in NC6-ID12F, was evidence of epistasis. If epistasis were absent, the magnitude of the effect of recombination would have been similar in each cross. Fitness loss in NC6-ID12F should be matched by an equal fitness gain in ID12-NC6F, but this was not observed. At least one recombinant differs significantly from expected fitness in three of the four tests, and ID12-NC6F had lower fitness than expected under each null hypothesis except that which assumes that gene F contributed nothing to overall fitness (and where recombination at F would thus have no effect on fitness). Additionally, both ancestors plateaued at different fitnesses when they were pre-adapted for optimal growth rate, implying that epistasis is at work within their genomes. The asymmetry of fitness effects showed that epistasis can interfere with successful hybridization even between individuals of very closely related populations. This may have implications for evolving populations, as even low levels of divergence can lead to hybrid incompatibilities that may reinforce isolation.

## Hybrid adaptation

We performed serial flask transfers of the ID12-NC6F and NC6-ID12F genotypes to allow for adaptation following potential disruptions of intergenic epistatic interactions resulting from recombination. In the recovery of ID2/ID12 hybrids previously performed by *Rokyta & Wichman (2009)*, the first mutation to fix in each recovery population was responsible for an average fitness increase of 6.8 doublings per hour and represented 60% of fitness recovery (*Rokyta & Wichman, 2009*). For this reason, we halted flask passaging of each recovery population after the fixation of the first mutation. Flask passaging of ID12-NC6F was halted after an apparent increase in fitness was observed following passage 16, indicating the likely fixation of a new mutation. Passaging of NC6-ID12F was similarly halted after 20 flasks, or 60 generations.

We analyzed the final adapted populations (ID12-NC6FRec and NC6-ID12FRec) by means of whole genome sequencing. We found that an A → G mutation at genome

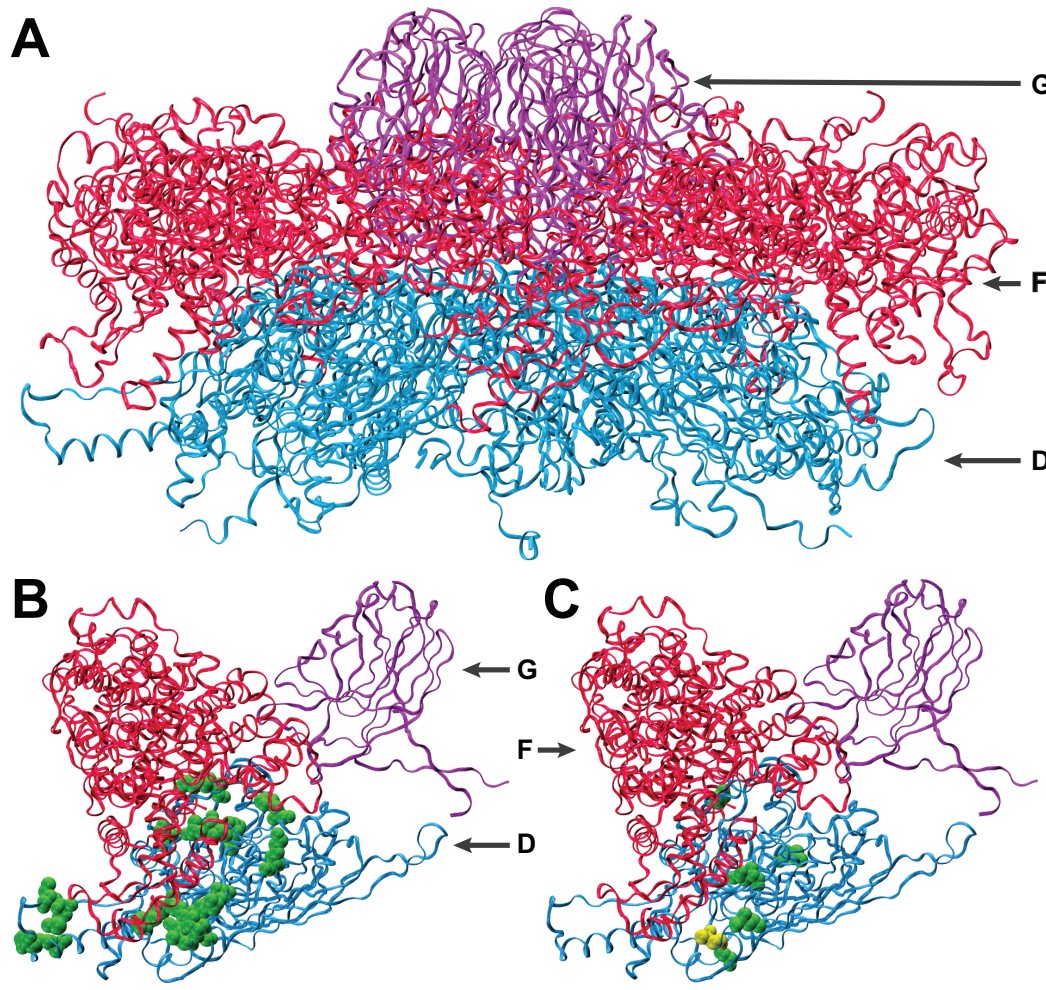

**Figure 2 Interactions of coat protein F with the external scaffolding protein D.** (A) An assembled pentamer of coat protein F (blue) with associated major spike proteins G (violet) and scaffolding proteins D (red). There are four copies of the scaffolding protein for each copy of the spike and coat proteins. (B) Single copies of the coat and major spike proteins with four copies of protein D. Twenty-two known sites of interaction between F and D are highlighted (green). (C) Sites of amino acid divergence between ID12 and NC6 are highlighted (green) as well as the site of the mutation that fixed during ID12-NC6F recovery (yellow). All five sites of ID12-NC6 divergence are located near known sites of D–F interaction, and the location of the recovery mutation at an interaction site strongly indicates that disrupted D–F interactions during procapsid formation are responsible for the observed epistatic effects on growth rate.

position 3485, resulting in a T → A amino acid substitution at amino acid residue 311 in the coat protein F, had fixed in the recovery population of ID12-NC6 (Fig. 2). The fitness of an isolate from the ID12-NC6FRec population (confirmed to contain only the mutation at 3485) was measured to be 24.3 ± 0.29, which was not significantly different from the fitness of ID12-NC6F (*t*-test, two-sided, unequal variance, $P = 0.24$; Fig. 1) or the fitness of the ID12 ancestor (*t*-test, two-sided, unequal variance, $P = 0.11$; Fig. 1). A single substitution (T → C) fixed in the NC6-ID12F recovery line at genome position 3233, resulting in an S → P amino acid substitution at residue 228 in the coat protein. The fitness of an isolate from the NC6-ID12FRec population (confirmed to contain only the mutation

at 3233) was measured to be $25.7 \pm 0.37$, which was not significantly different from the fitness of NC6-ID12F ($t$-test, two-sided, unequal variance, $P = 0.74$; Fig. 1). These results presented a stark contrast to those observed by *Rokyta & Wichman (2009)* in ID2/ID12 hybrids, where recombinants attained large fitness gains and the first mutation accounted for 60% of fitness recovery. This contrast was likely a result of the low level of divergence between the two ancestors and the relaxed epistatic costs of recombination which resulted in lower potential for fitness recovery.

Both recovery mutations fixed rapidly in populations of relatively large size, each initiated by a single clone, and no mutations were observed in the sequence of the population from a control line of the already well-adapted ID12 ancestor after 16 growth periods. In addition, no mutations fixed in a control line of the NC6 ancestor after 20 growth periods. We can therefore assume that these recovery mutations are beneficial, albeit with small effect sizes that were not detectable with our assay method. In addition to being assumed to be beneficial in its hybrid context, the mutation at residue 311 in the coat protein F which fixed in the ID12-NC6F population is located at a known site of contact between the coat protein and the external scaffolding protein (D) during the procapsid stage of assembly (*Dokland et al., 1999*), as discussed further in the next section (Fig. 2). For these reasons, we created the novel genotype ID12Mut by site-directed mutagenesis by substituting this recovery mutation into the ancestral ID12 genome to assess whether this mutation at a site critical to correct assembly dynamics between the coat protein and scaffolding protein was only beneficial in the context of the novel F allele. The fitness of the ID12Mut genotype was determined to be $23.5 \pm 0.30$, not significantly different from the fitness of the original ID12 ancestor without the mutation at position 3485 ($t$-test, one-sided, unequal variance, $P = 0.85$; Fig. 1). Although the mutation did not result in a significant fitness increase in ID12-NC6F, it did fix in the hybrid population, and it does not appear to be beneficial in the ancestor and did not appear in the control line of ID12 that was grown for 16 flask transfers, or 48 generations. That this mutation is beneficial only in the context of the hybrid background was indicative of epistasis governing the mutation's effect.

## Interactions with scaffolding proteins

The first step of viral capsid assembly for microvirids is the formation of procapsid subunits consisting of pentamers of the coat (F) and major spike (G) proteins. The F and G pentamers bind in a reaction mediated by the B and D scaffolding proteins to form a complete procapsid structure. Procapsid formation is mediated by 240 copies of protein D which form an external shell around the structure, with four copies of protein D in contact with each F protein (*Dokland et al., 1999*; *Uchiyama & Fane, 2005*). The five amino acid sites that differ between the ID12 and NC6 versions of protein F were all located near sites where the coat protein interacts with the external scaffolding protein D (Fig. 2), and the $\alpha$-carbon of each amino acid was on average located only 15.2 Å from the nearest $\alpha$-carbon of protein D. The mutation observed in the ID12-NC6F recovery line was located at a known site of contact between the coat protein and the external scaffolding

protein (*Dokland et al., 1999*). Based on these data, it is likely that the fitness cost associated with hybridization at gene F in NC6-ID12F was a result of disruption of interactions between the coat protein and scaffolding protein during procapsid assembly. The NC6 and ID12 alleles of gene D differed at only two amino acid sites, 118 and 148, of the 152 composing the protein. Residue 148 on the external scaffolding protein D is located very close to the sites of the coat protein F where NC6 and ID12 differ, and this site is located approximately 8.7 Å from the site of the mutation that fixed in the ID12-NC6F recovery line. The recovery mutation at amino acid site 311 in the coat protein likely corrected disrupted protein interactions resulting from recombination. The conformational details of the amino acid sites involved in ID12/NC6 hybridization explained at least some of the intergenic interactions that are likely affecting fitness of hybrids with an exchanged coat protein gene.

## Conclusions

Hybridization of closely related ID12 and NC6 at gene F did not have nearly as a high a fitness cost as *Rokyta & Wichman (2009)* observed from a reciprocal cross of highly divergent phage relatives ID12 and ID2 at the same locus. Epistasis was clearly demonstrated in the experiment of *Rokyta & Wichman (2009)*, and we expected that homologous recombination at gene F would be less disruptive of intergenic epistatic interactions in more closely related ancestors. This result was observed in the hybrids of NC6 and ID12, but epistasis was nonetheless detectable. The measured fitnesses of ID12-NC6F and NC6-ID12F deviate significantly from several null fitness expectations, and the asymmetrical fitness effects of recombination are strong evidence that epistasis is present in these genomes despite low levels of sequence divergence. The conditional benefit of the mutation observed during recovery of ID12-NC6F provides further indication that epistatic interactions are governing the fitnesses of these hybrids. It is likely that recombination at gene F may disrupt interactions between the coat protein and scaffolding protein D, and the location of the compensatory mutation in ID12-NC6FRec at a known site of interaction between proteins F and D may elucidate some facet of the epistatic interactions at work in this system (*Dokland et al., 1999*). Although the epistatic cost of hybridization between ID12 and NC6 was relatively small compared with the cost in more divergent phages, we showed that intergenic interactions can be disrupted by the exchange of gene homologs differing at even as few as five amino acid sites. We were successfully able to detect the effects of epistasis in very closely related bacteriophages and uncovered at least some part of the mechanisms of epistasis in this phage pair, and it follows from our results that if it is at all possible for divergent phages to yield hybrids with a total absence of epistatic costs, the parental genotypes in such a case must display exceptionally low divergence.

## ACKNOWLEDGEMENT

The authors thank S. Brian Caudle for laboratory assistance.

## Funding

Funding for this work was provided by the United States National Institutes of Health (NIH) to DRR (NIH R01 GM099723). The funders had no role in study design, data collection and analysis, decision to publish, or preparation of the manuscript.

## Grant Disclosures

The following grant information was disclosed by the authors:
United States National Institutes of Health (NIH): NIH R01 GM099723.

## Competing Interests

The authors declare there are no competing interests.

## Author Contributions

- Andrew M. Sackman performed the experiments, analyzed the data, wrote the paper, prepared figures and/or tables, reviewed drafts of the paper.
- Danielle Reed performed the experiments, reviewed drafts of the paper.
- Darin R. Rokyta conceived and designed the experiments, reviewed drafts of the paper.

## DNA Deposition

The following information was supplied regarding the deposition of DNA sequences:
The GenBank accession numbers of the original isolates are DQ079905 (ID12) and DQ079907 (NC6).

## Supplemental Information

Supplemental information for this article can be found online at http://dx.doi.org/10.7717/peerj.1320#supplemental-information.

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
