# Peer review of "Intergenic incompatibilities reduce fitness in hybrids of extremely closely related bacteriophages"

_PeerJ, doi:10.7717/peerj.1320_

## Round 0.1 · original submission · Minor Revisions

· Academic Editor

Minor Revisions

All three reviewers concurred that this is a well-done study, and most of the comments have to do with the clarity of the manuscript. I think this manuscript should be suitable for publication after the reviewer's comments are addressed. I would also urge the authors to have the manuscript proofread by someone else for clarity.

Reviewer 1 ·

Basic reporting

Excellent

Experimental design

Excellent

Validity of the findings

Seem totally valid

Additional comments

This is a really interesting, well written paper describing some well thought our experimental work on the fitness impacts of recombination in microvruses. I like it a lot and have no changes to suggest. I would be very interested to see in the future whether, besides compensatory mutation events, bacteria coinfected with more complex reciprocal chimaeras than those used here are able to produce fitter genomes through compensatory recombination. It may be possible to accurately identify large numbers of interacting sites by simply looking at patterns of sites that tend to be coinherited from the original wt viruses.

·

Basic reporting

No substantial comments. The raw fitness data are provided. The other data that might be included are the sequencing data used to identify the mutations, but those data are probably Sanger sequences of just a few thousand bases, and all relevant findings are in the paper. So there is likely nothing gained by providing or depositing those.

Experimental design

The study follows previously published work of the same nature. The difference here lies only in the strains used and in their similarities/divergence. The work is thus straightforward and easy to follow.

Validity of the findings

The interpretation of the data seems straightforward. I have a few quibbles about stats. First, lines 8-11 under Results and Discussion state: "Replacement of the allele for gene F in the ID12 genome by the homolog from NC6 resulted in a phage, ID12-NC6F, with fitness 23.7 ± 0.33, slightly higher than the fitness of its primary ancestor, ID12. The fitness of ID12-NC6F did not differ significantly from the fitness of its ID12 ancestor (t-test, two-sided, unequal variance, P = 0.50; Figure 1)." This is a bit misleading; better and shorter to say that the estimated fitness is slightly higher but not significantly so. Second, in calculating the expected fitnesses under the different "F contribution to fitness" models, I would expect those calculations to include standard errors (which may not be trivial to calculate). Note that what is really being considered is the contribution of variation in F to variation in fitness. It makes no sense to talk about the contribution to fitness of an essential gene.

Additional comments

Introduction:
Roger Hendrix is probably the person most responsible for our appreciation of recombination in phages. His work should be cited. There is also a famous 1970s paper by David Botstein in which he proposed modular evolution of phages by recombination.

There are very different issues between the evolution of recombinants in nature (where you never see the possibly 99.9% of recombinants that are bad) and artificial recombinants produced in the lab, in which you see the entire distribution of fitness effects. I would partition the Introduction accordingly.

A recent Science paper by Ed Marcotte's group did human gene substitutions in baker's yeast on a massive scale. Also, Paul Nurse won a Nobel prize from a gene substitution between fission yeast and humans. Those papers may (?) help provide a broad context for the study.

Results and Discussion:
The subheading "Hybrid incompatibilities evolve quickly" does not describe the contents of that subsection. I also think that all of the alternative epistasis models and calculations in that subsection could go in a table, so that the text could focus on higher level issues.

Figure 1 might benefit from a display of statistically significant differences within each cluster. Maybe this is not practical, however.

Reviewer 3 ·

Basic reporting

Sackman et al reciprocally swap a structural protein between two very similar bacteriophages, and find that there is a small but significant effect on fitness in one of the cases. The authors then evolve each recombinant phage and identify the first mutation to sweep in each case. These mutations have no measurable effect on growth rate, but are assumed to be beneficial because they swept to fixation. The mutation in one of the recombinants was determined to be in a region involved in protein-protein interactions.

I think that this manuscript would benefit from a more rigorous definition and testing of the epistasis. Condensing the manuscript would likely improve the clarity. It is confusing that figure 1 contains multiple bars which represent the exact same data.

Experimental design

The expectations for epistasis should be more quantitatively explained.

Validity of the findings

The data are robust, statistically sound and controlled.

---

## Round 0.2 · accepted · Accept

· Academic Editor

Accept

This manuscript received uncommonly positive comments from all three reviewers. They all agreed that this is an interesting study, and the authors have dealt with the few minor suggestions made during review.